# Long-Term Conservation Tillage and Precision Nutrient Management in Maize–Wheat Cropping System: Effect on Soil Properties, Crop Production, and Economics

Biswajit Pramanick [1,*], Mritunjay Kumar [1], Banavath Mahesh Naik [1,2], Mukesh Kumar [1], Santosh Kumar Singh [3], Sagar Maitra [4], B. S. S. S. Naik [5], Vishnu D. Rajput [6,*] and Tatiana Minkina [6]

1  Department of Agronomy, Rajendra Prasad Central Agricultural University, Pusa 848125, Bihar, India
2  Department of Agronomy, GB Pant University of Agriculture & Technology, Pantnagar 263145, Uttarakhand, India
3  Department of Soil Science, Rajendra Prasad Central Agricultural University, Pusa 848125, Bihar, India
4  Department of Agronomy, Centurion University of Technology & Management, Paralakhemundi 761211, Odisha, India
5  Department of Agronomy, ANGRAU, Bapatla 522101, Andhra Pradesh, India
6  Academy of Biology and Biotechnology, Southern Federal University, 344090 Rostov-on-Don, Russia
*  Correspondence: biswajit@rpcau.ac.in (B.P.); rajput.vishnu@gmail.com (V.D.R.)

**Abstract:** Intensive tillage coupled with imbalanced nutrient management in maize–wheat systems in low-carbon calcareous soils often results in poor productivity vis-à-vis degradation in soil health. Conservation tillage viz. permanent bed planting (PB) and zero tillage (ZT)/direct seeding with residue retention coupled with precision nutrient management might improve soil properties and yield of crops. Concerning this, a long-term experiment was conducted from 2014–2015 to 2020–2021 with a maize–wheat cropping system at TCA, Dholi farm of RPCAU, Pusa. Treatments consisted of three main plots of different tillage practices, viz. PB, ZT, and conventional tillage (CT) and three sub-plots of nutrient management options, viz. farmers' fertilization practice (FFP), site-specific nutrient management with Nutrient Expert® (NE) software, and GreenSeeker (GS) based nitrogen-management. From this study, it was observed that both the PB and ZT resulted in about 31–33% and 43–45% improvement in SOC and water-soluble aggregates (WSA), respectively, comparing them under CT. These two conservation tillage practices also improved the other soil bio-chemical properties. Better soil properties under PB and ZT helped in the improvement of system yield by about 13–18% comparing yield under CT. Moreover, both these tillage practices showed an additional net return of USD 330–USD 400 over CT. PB was found a bit better over ZT concerning soil properties, yield, and economics. Comparing nutrient management options, precision nutrition using NE and GS showed significant improvement in the soil bio-chemical parameters, yield, and economics of the cropping system over FFP. SSNM using NE showed slightly better results than GS. Thus, from this long-term study, it can be concluded that the permanent bed system with residue retention and precision nutrition using Nutrient Expert® software are the best options concerning tillage and nutrient management, respectively, for improvement of the soil properties of problematic calcareous soils, thereby, enhancing the yield and economics of the maize–wheat cropping system.

**Keywords:** maize–wheat system; permanent bed; zero tillage/direct seeding; smart crop nutrition; soil bio-physico-chemical properties; system yield and profits; calcareous soils

## 1. Introduction

Despite global efforts in recent decades to achieve food sufficiency for the world population, stagnation in the yield as well as a fall in the yield gain constituting about 31% of total global rice, wheat, and maize production [1,2] has been observed. Concomitantly, farmers are being increasingly necessitated to decrease the adverse environmental effects

that have been occurring over the years in the form of soil deterioration, groundwater contamination, and greenhouse gas emissions [3]. Soil properties, the internal factor, management strategies, and external factors are mainly influencing crop yields [4]. Agricultural scientists and farmers are left with the task of improving crop productivity by making the environment safe. Cereal-based cropping systems managed with conventional intensive practices have registered high production and energy costs, low input use efficiency, and environmental pollution. Conventional tillage systems which usually involve the inversion of soil have made the soil a source of greenhouse gases instead of a sink for greenhouse gases [5]. Fertilizers are an important source of nonpoint pollution from agriculture. Cropping practices such as tillage and crop fertilizer management are an important management tool for sustainable increasing crop yields. This necessitates a paradigm shift in cropping practices [6].

Minimum soil disturbance, plus soil cover with residue retention, coupled with a diversified sustainable cropping system are the basics of conservation agriculture [3]. Vanlauwe et al. [7] proposed and embraced the 4th pillar of conservation agriculture, i.e., appropriate use of fertilizers. Thus, conservation agriculture systems must be divisible and flexible in nature allowing benefits under diverse situations [8]. In eastern India, maize appears to be one of the most emerging crops fore farmers, being suitably fitted in rice-based systems [9]. Srinivasan et al. [10] suggested that enhancing maize production is needed for the hour to meet the rising demand. In the calcareous soils of eastern India, maize area is increasing rapidly and farmers are replacing rice with maize crops having numerous benefits. Thus, the maize–wheat system has become a very promising one [11]. The main production constraints in northeastern Indo-Gangetic plains (IGP) are high farm holding, poor marketing infrastructure, low technology adoption, poor infrastructure, and aberrant climatic events such as hailstorms, flooding, and heat stress. Maize systems with conservation management practices have recorded benefits such as enhancing system productivity, soil health, environmental quality, and saving water [12,13].

The calcareous soils of the eastern part of India have been facing several challenges due to continuous intensive tillage practices along with imbalanced fertilization leading to several nutrient deficiencies and loss of soil organic carbon (SOC) [14]. Conservation tillage such as permanent bed formation, and zero tillage with residue retention can mitigate this problem by improving soil bio-physico-chemical properties [15] which ultimately enhance the yield of crops [16]. The conservation-based maize system provides resilience to erratic rainfall conditions in eastern IGP [9]. A permanent raised-bed system is very helpful for crops that cannot tolerate water logging [17]. Generally, the rainfall amount is higher in the eastern part of India and the land situation is medium, which might result in water logging from the crops. Thus, permanent-bed planting was adopted in this study. Another important aspect of the maize–wheat cropping system is balanced nutrition, particularly in the calcareous soil where several nutrient deficiencies are observed. Maize growers of eastern India mainly emphasize applying nitrogenous fertilizer without considering the balanced nutrition leading to poor nitrogen-use-efficiency (NUE) [18]. Site-specific nutrient management (SSNM) using Nutrient Expert® software can improve the crop yield by providing nutrition to the crop at the right time with the right dose [18]. GreenSeeker-based nitrogen (N)-management is also an important fertilization practice that helped in proper N-nutrition of the crop, improving the overall NUE [19,20].

All these above-mentioned facts represent that the conservation tillage practice coupled with precision nutrient management can improve soil health and system yield of maize–wheat cropping systems. There are good kinds of literature available on soil health and crop yield as influenced by conservation tillage. However, the scientific literature on the impact of conservation tillage coupled with precision nutrient management on soil health of problematic calcareous soils of eastern India, productivity, and economics of promising maize–wheat cropping system are scarce. Thus, the major aim of the present long-term study on conservation tillage with different nutrient management options was to find out the impact of these conservation agricultural practices on varied soil parameters

(physical, chemical, and biological), system yield, and economics of maize–wheat cropping system grown under low organic carbon calcareous soils of eastern India. This study hypothesized that the conservation tillage along with precision nutrition could enhance the soil properties, crop production, and economics of maize–wheat system. The novelty of the present study is to generate novel information on the impact of conservation tillage and precision nutrient management options on soil health, system yield, and economic benefits of the maize–wheat system in problematic calcareous soils.

## 2. Materials and Methods

### 2.1. Place of Study

The study was performed at the TCA, Dholi farm of Dr. Rajendra Prasad CAU, Pusa, Bihar, India (latitude: 25°59′ N, longitude: 85°40′ E, altitude: 52.9 m above mean sea level) for 7 years from 2014–2015 to 2020–2021. The soil of the experimental site was sandy-loam in texture which falls under the soil taxonomical class, typic calciorthent. Soil pH was alkaline in reaction with the value of 8.7, while the free $CaCO_3$ of the soil was 33.2%. The other initial nutrient status of the study soil (0–15 cm depth) before initiation of this long-term trial is presented in Table 1. The study area falls under a sub-tropical humid climate with mean annual rainfall and evaporation of 1150 mm and 170 mm, respectively. The rainfall pattern and evaporation during the 7 years of study are presented in Figure 1. The mean minimum temperature varied from 6.0–7.2 °C during January, while the mean maximum temperature varied from 36.5–37.8 °C during May. Values of relative humidity (RH) reached up to 98% during July–August, while the RH value fell to 35% in the month of March.

**Table 1.** The initial status of the soil properties (at 0–15 cm depth) of the study site before sowing the rainy season maize during 2014.

| Attributes | Values |
| --- | --- |
| Soil physical properties | |
| Bulk density (BD) | 1.38 Mg m$^{-3}$ |
| Water soluble aggregates (WSA) | 44.7% |
| Water holding capacity | 29.0% |
| Soil chemical properties | |
| Soil organic carbon (SOC) | 4.8 g kg$^{-1}$ of soil |
| Available nitrogen (N) | 242 kg ha$^{-1}$ |
| Available phosphorus (P) | 14.3 kg ha$^{-1}$ |
| Available potassium (K) | 118 kg ha$^{-1}$ |
| Zinc (Zn) content | 0.85 ppm |
| Iron (Fe) content | 8.45 ppm |
| Soil biological properties | |
| Active carbon (AC) | 35.8 mg kg$^{-1}$ soil |
| Microbial biomass carbon (MBC) | 402 µg C g$^{-1}$ soil |
| Dehydrogenase activity (DHA) | 23.5 µg TPF Rel g$^{-1}$ day$^{-1}$ |

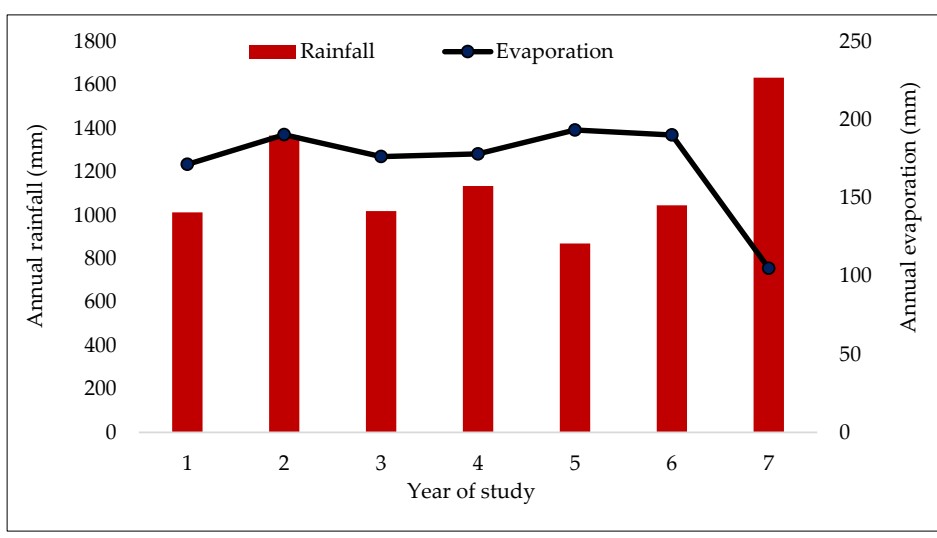

**Figure 1.** Annual rainfall and evaporation during experimental years.

*2.2. Experimental Details*

Statistical design split-plot design (SPD) was followed for this study with 3 replications. The main plots consisted of different tillage practices such as permanent bed system (PB), zero-tillage system (ZT)/direct seeding, and conventional tillage (CT). 3 sub-plots containing varied nutrient management practices, viz. farmers' practice (FFP), site-specific nutrient management (SSNM) using Nutrient Expert® software, and GreenSeeker-based nutrient management were superimposed in each main plot. The size of each subplots was 5 m × 5.4 m. The study started in the rainy season of 2014 with maize. The cropping system for this study was maize–wheat, maize in the rainy season, and wheat in the winter season. Before the initiation of the study, the layout was made and kept permanent for the entire experimentation period. Under CT, the field was cross-ploughed with disk-harrow, followed by cultivation with rotavator and planking before sowing of each crop. Under ZT plots, crops were directly sown using ZT-planter. In the ZT/direct seeding plots, at least 30% of the residues of the previous crop were maintained in the field. In the case of the PB system, fresh beds of 37 cm flat-top with 15 cm height were made. The distance between the two beds from their middle point was 67 cm. Beds were kept permanent and reshaping was performed using a disk-coulter before sowing each crop. At least 30% of crop residues were also kept in permanent beds as in ZT plots. The number of nutrients applied in the FFP, SSNM, and GreenSeeker-based practices are presented in Table 2.

**Table 2.** Description of the treatments.

| Treatments | Maize | Wheat |
|---|---|---|
| *Tillage practices* | | |
| Permanent bed system (PB) | Permanent bed of 37 cm flat-top with 15 cm height and 67 cm apart from the middle point of two beds. At least 30% of the residue of the previous crop was retained. | Same as maize |
| Zero tillage (ZT)/Direct seeding | Sowing was performed using ZT-planter. At least 30% of the residue of the previous crop was retained. | Same as maize |
| Conventional tillage (CT) | Two cross-harrowing followed by 1 pass of rotavator and planking. No residues of the previous crop were retained | Same as maize |

**Table 2.** *Cont.*

| Treatments | Maize | Wheat |
|---|---|---|
| *Nutrient management* | | |
| Farmers' practice (FFP) | 155, 60, and 40 kg ha$^{-1}$ of N, P, and K, respectively | 120, 55, and 40 kg ha$^{-1}$ of N, P, and K, respectively |
| Site-specific nutrient management using Nutrient Expert software (SSNM) | 140, 64, and 45 kg ha$^{-1}$ of N, P, and K, respectively | 122, 63, and 42 kg ha$^{-1}$ of N, P, and K, respectively |
| GreenSeeker-based nutrient management | 132 kg ha$^{-1}$ of N; 60, and 40 kg ha$^{-1}$ of P and K, respectively | 115 kg ha$^{-1}$ of N; 60, and 40 kg ha$^{-1}$ of P and K, respectively |

*2.3. Crop Establishment*

The most farmer-adapted varieties of maize (variety, Shaktiman 5) and wheat (variety, HD 2967) of the experimental region were chosen for this experiment. In the first year of the study, 1.5 Mg ha$^{-1}$ of wheat straw was kept in ZT/direct seeding and PB plots, and from the next season onwards, no less than 30% of the residues of the previous crop were kept in the ZT/direct seeding and PB plots. No residues were kept in the case of CT plots. Maize was sown in mid-June and harvested at end of September for each year of the study, while wheat was sown from the end of October to the first week of November and harvested during mid-April. Under the conservation plots (both PB and ZT/direct seeding), weeds were killed by spraying Glyphosate at 1.5 kg ha$^{-1}$ before 15 days of sowing the maize and wheat. The seed rate for maize and wheat was 20 kg ha$^{-1}$ and 100 kg ha$^{-1}$, respectively. Maize seeds were sown at a row–row spacing of 67 cm and plant–plant spacing of 15–20 cm, while wheat seeds were sown with 22.5 cm with row-to-row spacing. Weeds of the field were managed using chemical weeding two times for both crops. For weed control in maize, pre-emergence application of Atrazine at 1000 g ha$^{-1}$ at 2 days after sowing (DAS) followed by 1 hand weeding during 30 DAS were practiced, while in the case of wheat, PRE application of Pendimethalin at 1 kg ha$^{-1}$ at 2 DAS followed by 2,4-D Na-salt at 500 g ha$^{-1}$ during 30 DAS was followed. Concerning nutrient management, FFP was selected by surveying 50 maize and wheat growing farmers of the study region. Under FFP, 50% of the N and entire P and K nutrients were applied as basal for both crops. The rest 50% of N was top-dressed twice for maize during the knee-height and tasselling stage, and for wheat during the active tillering and pre-flowering stage. Under SSNM, IPNI developed a software-based decision support system, Nutrient Expert® was used to fix the target yield of the crops. SSNM-based nutrient application rates for maize and wheat were 140:64:45 and 122:63:42 kg ha$^{-1}$ of N, P, and K, respectively. Calculated full doses of P, 1/3rd N, and 2/3rd K were applied during land preparation/sowing. The remaining 2/3rd N were top-dressed at an equal split during the 8th leaf and tasselling stage of maize, and the active tillering and spikelet formation stage of wheat. The remaining 1/3rd of K was applied at the grain-filling stage of wheat and maize. In the case of GreenSeeker-based treatment, P and K-nutrients were managed as mentioned in the case of SSNM. However, the N was managed using the hand-held NDVI-sensor, GreenSeeker reading (Trimble Inc., Sunnyvale, CA, USA). GreenSeeker reading indicated whether the plant needs nitrogen or not. The reading for both the crops was taken after 3 weeks of sowing and as per the GreenSeeker reading, a total of 132 kg ha$^{-1}$ N was applied in maize in 4 splits, while the total N-application amount for wheat was 115 kg ha$^{-1}$ with 3 splits. In the case of maize, two irrigations were given during the knee-height and tasselling stage, while in the case of wheat, 3 irrigations were given during the crown–root initiation, active tillering, and spikelet formation stages.

*2.4. Soil Sampling and Analysis*

Initial soil samples were taken before the commencement of the study in 2014, with soil samples again collected after the completion of 7 years of the study in 2021. Soil samples were collected using a soil auger from 0–15 cm soil depth following standard protocol.

Afterward, samples were air-dried and processed to analyze the soil physical, chemical, and biological parameters.

### 2.4.1. Soil Physical Properties

A core sampler was used for bulk density (BD) determination. Soil samples from 0–15 cm depths were taken with a core sampler followed by oven-drying at 105 °C for 24 h. Oven-dried soil weight was divided by the known volume of the core to determine the BD as mentioned in Blake and Hartge [21]. The wet-sieving method was used to determine the water-soluble aggregates in soil with the help of the Yodder's apparatus [22]. To determine this, 50 g of processed soil samples were passed through an 8 mm sieve, and then transferred into different diameter sieves of 4 mm, 2 mm, 1 mm, 0.5 mm, 0.25 mm, and 0.125 mm. Afterward, the sieved soils were soaked with water for 10 min. followed by hand shaking for 10 min. to get grouped into different aggregate classes. The water-holding capacity of the soil was determined using the gravimetric method. Soils were collected in the known weight, uniform volume soil–moisture boxes, and the moist–soil weight was taken; afterwards, the soils in the moisture box were oven dried at $105 \pm 2$ °C. Then, the dry weight was taken to know the weight of the moisture kept in the soil.

### 2.4.2. Soil Chemical Properties

The Walkley and Black [23] method was followed to determine SOC by a rapid-titration method using potassium dichromate, orthophosphoric acid, sodium fluoride, and sulphuric acid. An amount of 0.5 g processed soil, 10 mL potassium dichromate, and 20 mL concentrated sulphuric acid were mixed well in a comical flask and kept for 30 min. Then, 250 mL of distilled water, 10 mL of orthophosphoric acid, and a pinch of sodium fluoride were mixed and allowed to get cooled. Afterward, 1 mL of diphenylamine indicator was mixed and titrated with ferrous ammonium sulphate to find out the SOC. Determination of soil available, N, was carried out following the Modified Kjeldahl method as described by Jackson [24], while the available phosphorus was estimated by Olsen's method (Jackson, 1973) [24] with the help of a UV-VIS double beam spectrophotometer (Systronics India Pvt. Ltd., Ahmedabad, India). Soil available potassium was determined with the help of a Flame photometer (Systronics India Pvt. Ltd., Ahmedabad, India) after soil extraction with 1 (N) ammonium acetate [24]. Atomic absorption spectrophotometer (Perkin Elmer, Waltham, MA, USA) was used to determine Zn and Fe content in soil [25]. The detailed procedure was mentioned in our previously published paper [14].

### 2.4.3. Soil Biological Properties

Active carbon was measured calorimetrically based on the protocol adapted from Weil et al. [26]. Air-dried and 2 mm sieved, 2.5 g soil sample was placed in a 50 mL centrifuge tube. The soil was added with 18 mL distilled water + 2 mL 0.02 M $KMnO_4$. Afterward, 2 min. Handshaking of the tubes was performed to make the active carbon oxidize in the sample. Then the tubes were kept for 10 min. for settlements and 0.2 mL supernatants were collected in 20 mL distilled water. Different volumes of standard dilutions of $KMnO_4$ were made to make the standard curve after checking the absorbance at 550 nm wavelength. Then, the absorbance of the sample was measured and compared with the standard curve. Finally, the sample absorbance was converted to active carbon in units of mg kg$^{-1}$ of soil. Soil microbial biomass carbon (MBC) was determined following the methods mentioned by Vance et al. [27]. The details are mentioned in our previously published paper [28]. For estimating the dehydrogenase activity (DHA) in soil, the procedure described by Casida et al. [29] was followed. For this, 0.2 mL of 2, 3, and 5 triphenyl tetrazolium chloride was added with 1 g air-dried soil samples, and incubated for 1 day at $30 \pm 1$ °C. Afterward, methanol (10 mL) was added for hand-shaking for 1 min. Then, the colour intensity measurement was performed by a UV-VIS double beam spectrophotometer (Systronics India Pvt. Ltd., Ahmedabad, India) at 485 nm wavelength for determining the soil DHA activity.

*2.5. System Yield*

System yield was measured in terms of maize equivalent yield (MEY). Wheat yield was converted into MEY by using the following formula [30]:

MEY of wheat = (Wheat yield * unit price of wheat)/(The unit price of maize)

The total system yield was determined by adding the maize yield and MEY of wheat. The system yield of the 7th year is presented in this manuscript.

*2.6. Economics*

The following formula was used to estimate net return (in USD): Net return (USD) = Gross return (US$)–Cost of cultivation (US$); Benefit–cost ratio (B–C ratio) was calculated using the following formula [6]:

B-C ratio = (Gross income (US$))/(Cost of cultivation (US$))

In Supplemental Table S1, details of the cost of cultivation are given.

*2.7. Statistical Analysis*

All the data collected from the experiment were analyzed using two-way ANOVA. Data analysis was conducted using the statistical software package SAS v9.3 (SAS Inc., Cary, NC, USA). The treatment means were compared using the least significant difference (LSD) at $p = 0.05$ as described in Gomez and Gomez [31]. Correlation matrix were prepared using SPSS v24.0 (IBM Inc., Armonk, NY, USA). Software, SigmaPlot v.9.0 (Systat Software Inc., Chicago, IL, USA) was used to prepare the graphs presented in this research paper.

## 3. Results

*3.1. Soil Physical Properties*

Soil physical properties were significantly influenced by the different tillage practices for 7 years (Figure 2a–c). However, the long-term varied nutrient management viz. FFP, SSNM, and GreenSeeker-based N-management did not reflect wide variation in the soil's physical properties. The use of the permanent bed (PB) establishment method for maize–wheat cropping system for 7 continuous years resulted in a significantly higher number of water-soluble aggregates (WSA) in the soil compared to that under ZT/direct seeding, and CT (Figure 2a). This tillage system showed about 6% and 45% more amount of WSA (%) in the soil as compared to zero tillage/direct seeding, and conventional tillage systems, respectively. Under different nutrient management practices, WSA (%) was not varied significantly. However, GreenSeeker-based N-management showed 4% and 6% more amount of WSA (%) in the soil as compared to WSA (%) in SSNM and FFP, respectively. Regarding BD and water holding capacity of the soil, both PB and ZT/direct seeding showed statistically at par results (Figure 2b–c). Soil WHC (water holding capacity) was found about 1% and 25% higher in ZT comparing PB and CT, respectively. BD of the soil was found to be the minimum under ZT and the maximum in CT. BD under ZT and PB were found almost similar. However, BD under CT was found to be about 5–6% more comparing BD under ZT and PB. BD and the water holding capacity of the soil were not altered significantly with varied nutrient management options. Precision nutrient management using SSNM and GreenSeeker showed slightly better results comparing FFP concerning BD and water-holding capability.

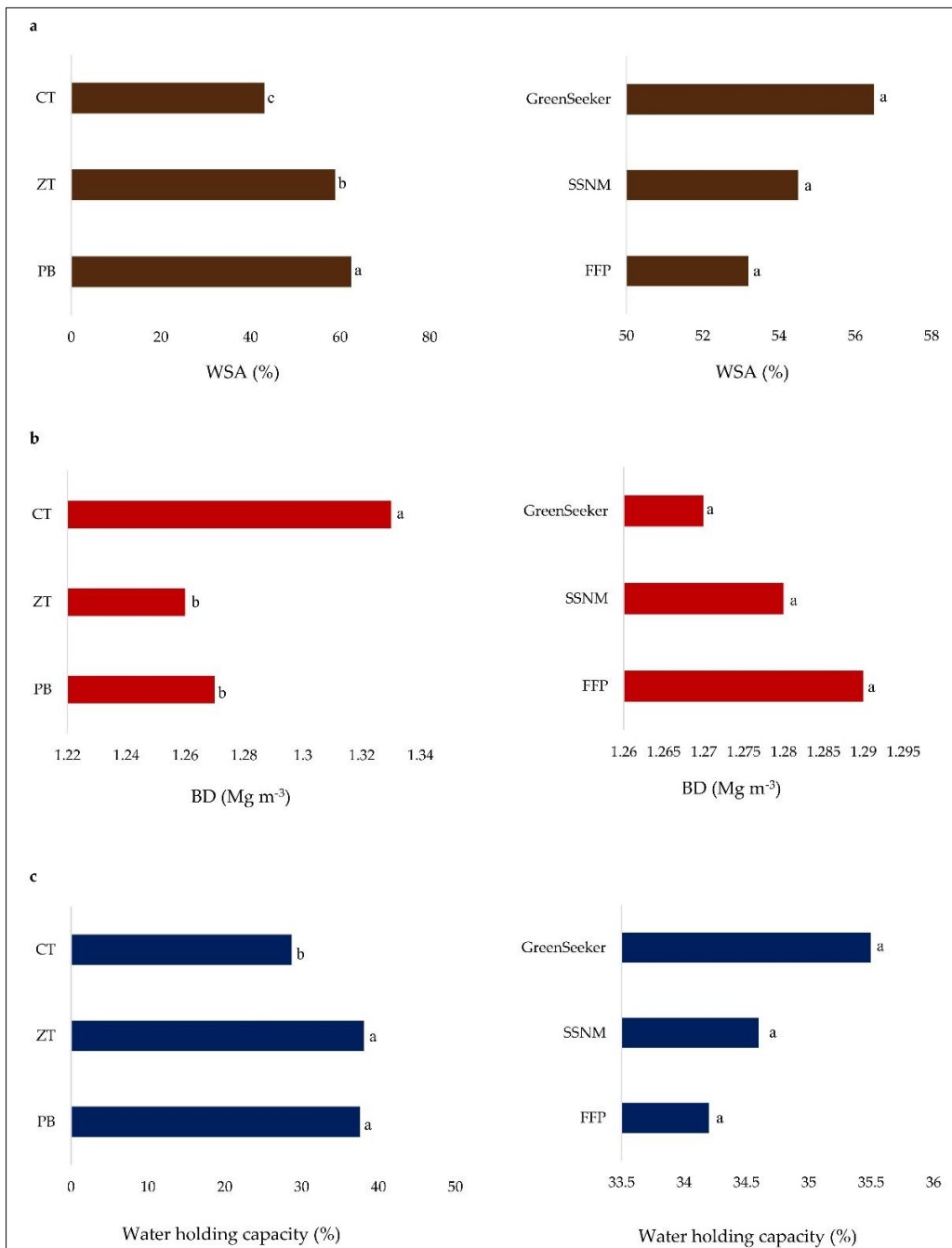

**Figure 2.** (**a**) Water soluble aggregates (WSA); (**b**) Bulk density (BD); (**c**) Water holding capacity under different tillage and nutrient management options [same letters after bars represent no statistical difference ($p \leq 0.05$); CT, ZT, and PB denote conventional tillage, zero tillage/direct seeding, and permanent bed, respectively; SSNM, and FFP denote site-specific nutrient management using Nutrient Expert Software, and farmers' practice, respectively].

### 3.2. Soil Chemical Properties

Soil organic carbon (SOC) was significantly influenced by the varied long-term tillage practices (Figure 3). The maximum amount of SOC was found in the permanent bed (PB) system being at par with zero tillage (ZT)/direct seeding practices. PB and ZT resulted in about 31–33% more SOC in the topsoil compared to that in conventional tillage (CT) systems after 7 years of study. Comparing the SOC in CT plots after 7 years with the initial SOC, it was also observed that there was a 1% decrease in the topsoil organic carbon in the plots where CT was applied. Under different nutrient management practices, there were no

statistical differences in the SOC content. However, the highest value of SOC was estimated under the GreenSeeker-based N-management plot comparing SSNM and farmers' practice (FFP). GreenSeeker-based N-management and SSNM resulted in 12.5% and 8.3% more SOC, respectively comparing the SOC under FFP.

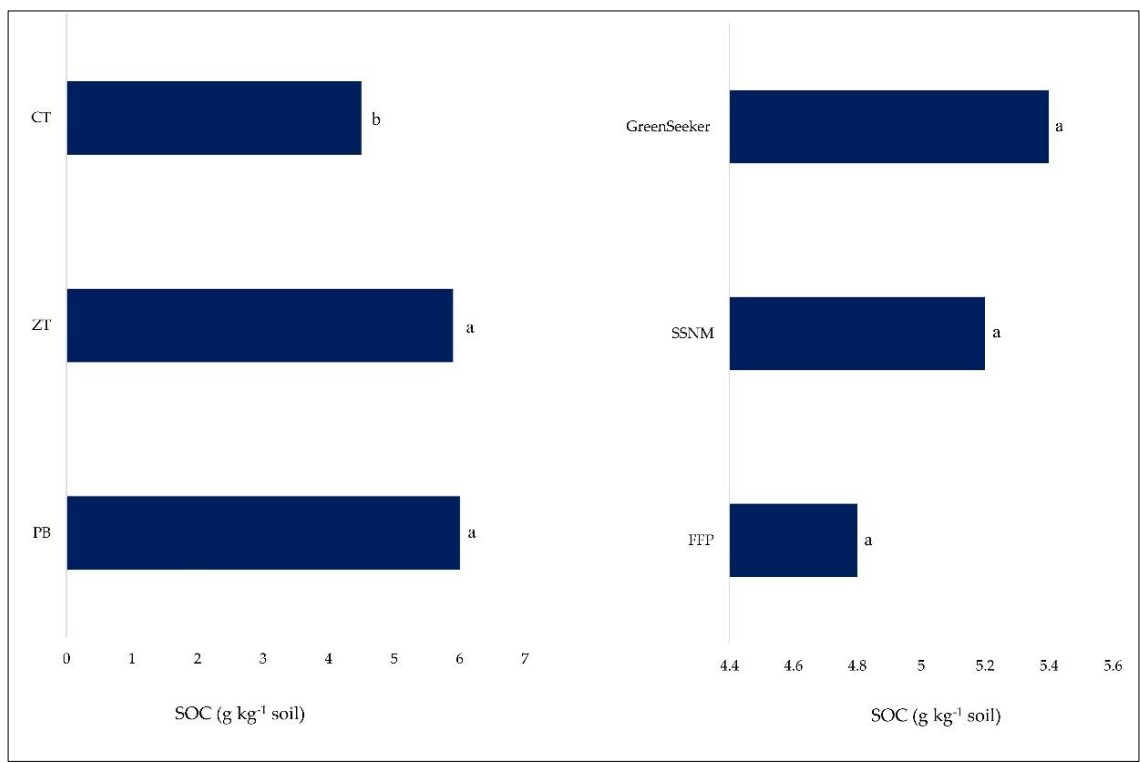

**Figure 3.** Soil organic carbon (SOC) under different long-term tillage practices and nutrient management options [same letters after bars represent no statistical difference ($p \leq 0.05$); CT, ZT, and PB denote conventional tillage, zero tillage/direct seeding, and permanent bed, respectively; SSNM, and FFP denote site-specific nutrient management using Nutrient Expert Software, and farmers' practice, respectively].

PB, ZT/direct seeding, and CT significantly alter all other soil chemical properties tested in this long-term study (Figure 4a–e). The highest attainments of soil available N, P, K, Zn, and Fe were found under PB practice being at par with ZT practice. PB and ZT practice showed 14–15%; 25–28%; 10–12%; 26–27%; and 6–8% more amount of soil available N, P, K, Zn, and Fe, respectively as compared to those under CT.

Concerning different nutrient management practices, it was observed that the soil available N and K were significantly influenced by different nutrient management practices (Figure 4a–e). However, other soil chemical parameters were not significantly influenced by different nutrient management. The highest amount of soil available N was observed in FFP, being statistically at par with SSNM. However, the available N in soil under FFP was only 3–5% higher as compared to the soil-available N under SSNM and GreenSeeker-based treatments. Soil available K was found to be the maximum under SSNM being at par with GreenSeeker-based treatment. SSNM practice resulted in 8% and 7% more soil-available K as compared to soil-available K in FFP and GreenSeeker-based treatments. Concerning other soil chemical properties, no such statistical differences among the treatments were observed. Soil available P, Zn, and Fe content in soil were found a little bit higher under SSNM comparing the other two nutrient management practices.

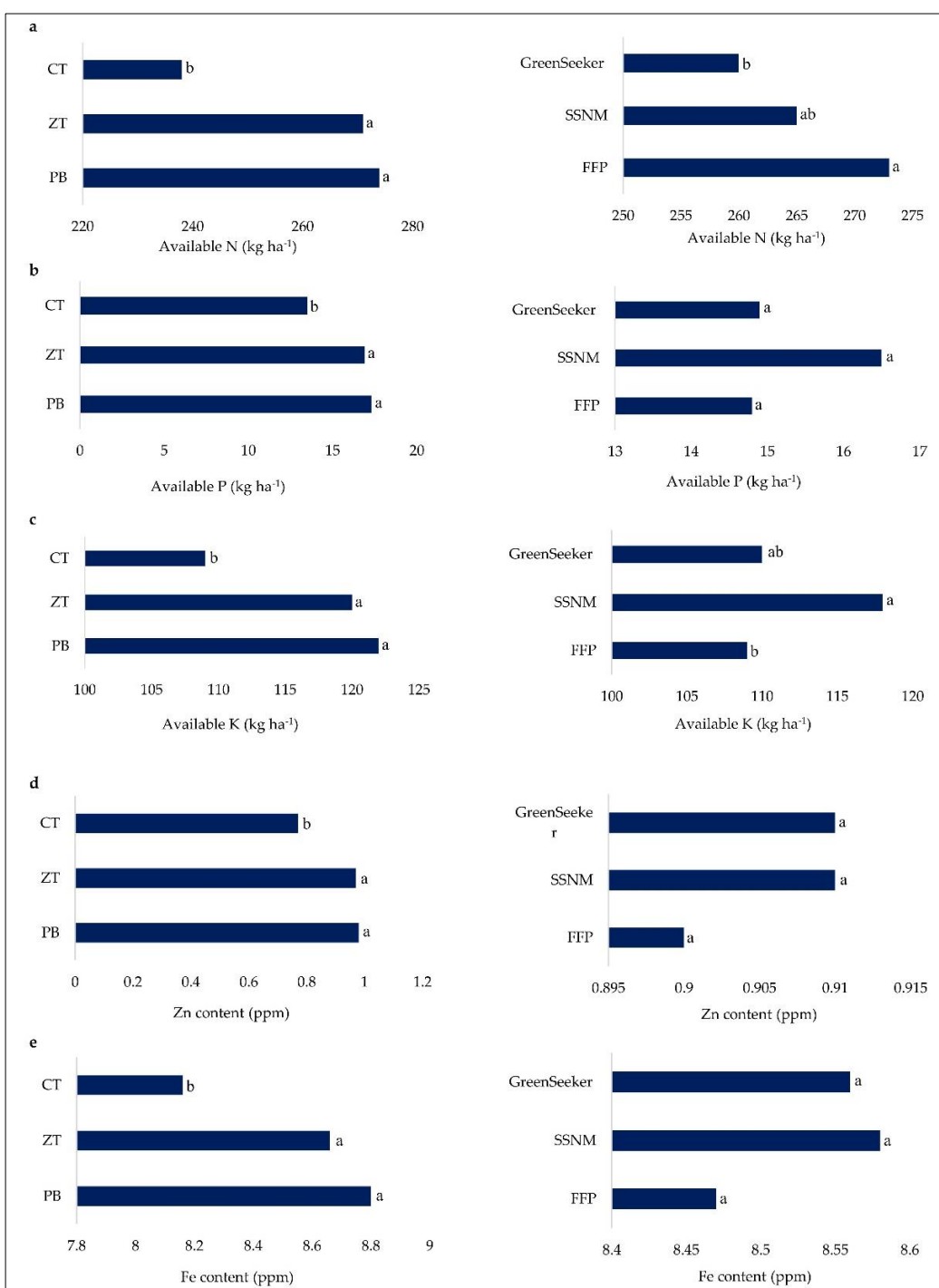

**Figure 4.** (**a**) Soil available N; (**b**) Soil available P; (**c**) Soil available K; (**d**) Zn content; (**e**) Fe content in soil under different tillage and nutrient management options [same letters after bars represent no statistical difference ($p \leq 0.05$); CT, ZT, and PB denote conventional tillage, zero tillage/direct seeding, and permanent bed, respectively; SSNM, and FFP denote site-specific nutrient management using Nutrient Expert Software, and farmers' practice, respectively].

### 3.3. Soil Biological Properties

Different tillage practices and nutrient management options significantly influenced the soil's biological properties (Table 3). Concerning PB, ZT/direct seeding, and CT practices, it was perceived that the best attainments of soil active C (AC), microbial biomass carbon (MBC), and dehydrogenase activity (DHA) were found in PB and this tillage prac-

tice was statistically at the same level with ZT/direct seeding. PB and ZT exhibited about 33–38%, 21–23%, and 41–49% higher values of AC, MBC, and DHA, respectively as compared to those under CT.

Concerning varied nutrient management options, precision nutrient management practices viz. SSNM and GreenSeeker-based N-management resulted in superior results concerning soil biological properties to the results found under FFP. AC was found to be the maximum under GreenSeeker-based management and this treatment showed about an 8% increase in the amount of AC comparing FFP, while the MBC and DHA were found at the maximum under SSNM and this treatment showed about 8% and 5% more amount of MBC and DHA, respectively, comparing FFP.

**Table 3.** Soil biological properties under different tillage and nutrient management practices.

| Treatments | Active Carbon (mg kg$^{-1}$ Soil) | Microbial Biomass Carbon (µg C g$^{-1}$ Soil) | Soil Dehydrogenase Activity (µg TPF Rel g$^{-1}$ Day$^{-1}$) |
|---|---|---|---|
| Tillage practices | | | |
| PB | 42.8 [a] | 440 [a] | 31.2 [a] |
| ZT | 41.1 [a] | 432 [a] | 29.5 [a] |
| CT | 31.0 [b] | 358 [b] | 21.0 [b] |
| LSD ($p \leq 0.05$) | 2.2 | 28 | 1.8 |
| Nutrient management | | | |
| FFP | 37.1 [b] | 390 [b] | 28.7 [b] |
| SSNM | 39.8 [b] | 420 [a] | 30.2 [a] |
| GreenSeeker based management | 40.1 [a] | 414 [a] | 30.0 [ab] |
| LSD ($p \leq 0.05$) | 2.8 | 20 | 1.3 |

The same letters after values are statistically at par ($p \leq 0.05$) [CT, ZT, and PB denote conventional tillage, zero tillage/direct seeding, and permanent bed, respectively; SSNM, and FFP denote site-specific nutrient management using Nutrient Expert Software, and farmers' practice, respectively].

### 3.4. System Yield and Economics

Results on system yield in terms of maize equivalent yield (MEY) after 7 years of the study are presented in Figure 5a. Concerning the tillage practices, the maximum MEY was recorded with PB being at par with ZT/direct seeding. Both PB and ZT resulted in 18% and 13% more MEY, respectively, comparing the MEY under CT.

Under varied nutrient management practices, the application of SSNM resulted in the maximum MEY being at par with GreenSeeker-based N-management (Figure 5a). SSNM and GreenSeeker-based management showed 12% and 10% more MEY, respectively, comparing MEY under FFP. Both the precision nutrient management practices ensured optimal nutrient supply to the crop as per their demand, resulting in better biomass production.

Concerning the economics, both the net return in terms of US$ and the benefit–cost ratio (B–C ratio) were significantly varied with different tillage practices (Figure 5b,c). The maximum values of the net returns and benefit–cost ratio was estimated under PB being at par with ZT. Conservation agriculture practice viz. PB and ZT showed an additional net return of about US$ 330–US$ 400 over the net return under CT. The cost of cultivation under PB and ZT is lower than the cultivation cost under CT as the land preparation cost is reduced under conservation tillage practices. Varied nutrition options were also found to influence the economics of the system (Figure 5b–c). The highest values of net returns and B–C ratio were evaluated with SSNM being at par with GreenSeeker-based nutrient management. SSNM exhibited 26% and 12% more net return and B–C ratio, as compared to those under FFP.

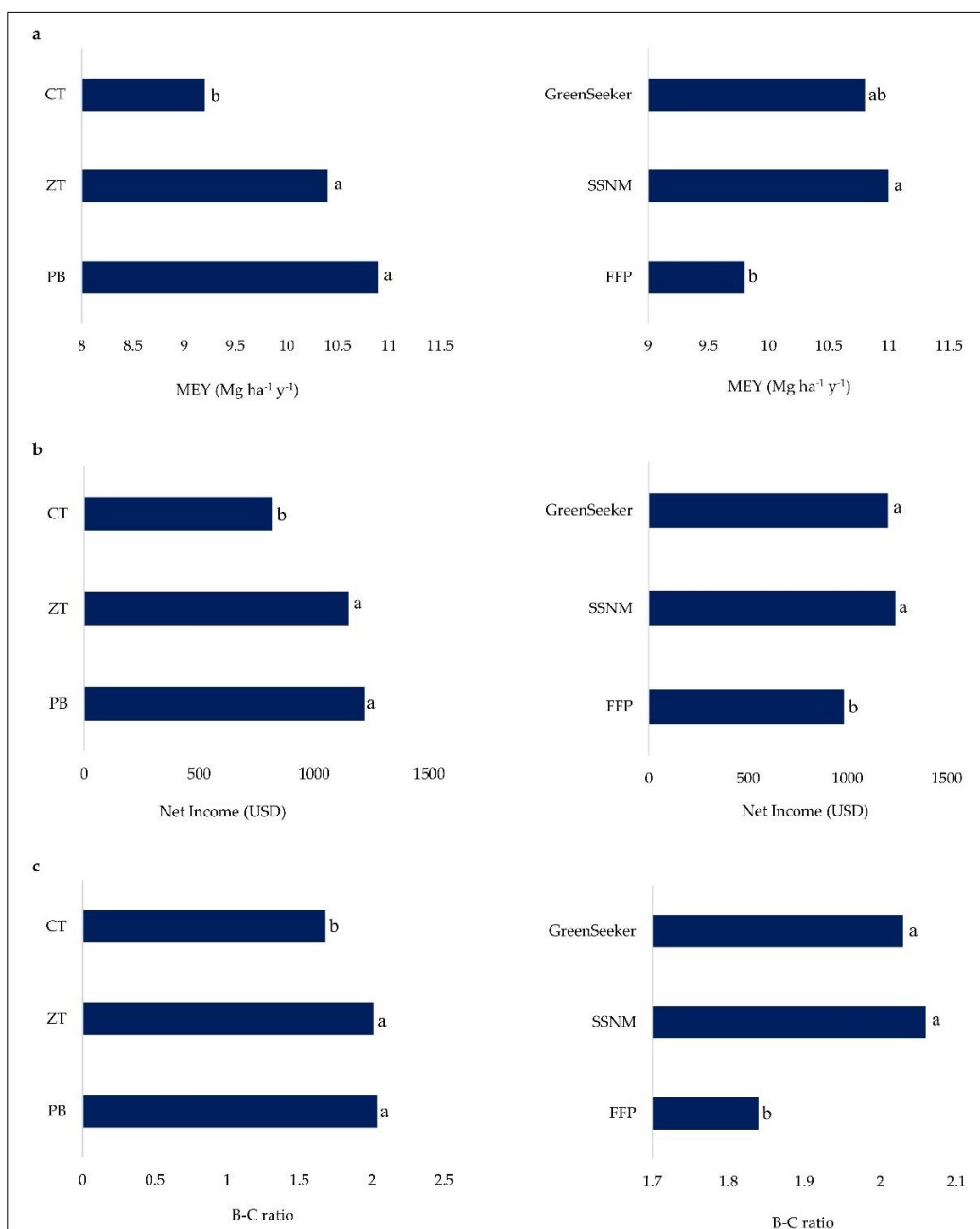

**Figure 5.** (**a**) Maize equivalent yield (MEY); (**b**) Net returns; (**c**) Benefit–cost ratio under different tillage and nutrient management options [same letters after bars represent no statistical difference ($p \leq 0.05$); CT, ZT, and PB denote conventional tillage, zero tillage/direct seeding, and permanent bed, respectively; SSNM, and FFP denote site-specific nutrient management using Nutrient Expert Software, and farmers' practice, respectively].

## 4. Discussion

Conservation agriculture practices viz. PB and ZT/direct seeding resulted in significant improvement of soil physical properties after 7 years of study. Both PB and ZT showed ~40–45% improvement in WSA (%) comparing WSA (%) in CT. Such improvement in the WSA (%) might be the result of higher SOC build-up in the conservation agriculture-based practices viz. PB and ZT over CT [32]. A strong relationship between WSA and SOC as found in Equation (1) also explained such findings.

$$\text{WSA} = 0.083\text{SOC} + 0.892;\ R^2 = 0.941 \tag{1}$$

Soil BD was found to be reduced under ZT/direct seeding and PB as compared to BD under CT. On the other hand, the water-holding capacity (%) of soil was improved under the conservation tillage practices compared to those under continuous conventional tillage (CT). Long-term conservation practices resulted in a higher amount of SOM build-up with better soil aggregation which finally resulted in lower BD and higher water-holding capability, as demonstrated by Salem et al. [33], Parihar et al. [15], and Kar et al. [3] from their earlier studies. A strong relationship between BD and SOC as well as WSA as found in Equations (2) and (3) also explained such findings.

$$BD = -20SOC + 31.33; R^2 = 0.929 \tag{2}$$

$$BD = -218.8WSA + 337.9; R^2 = 0.819 \tag{3}$$

Different nutrient management options were not found to impact the soil's physical properties significantly. However, precision nutrition using GreenSeeker-reading resulted in the reduction of BD and improvement of WSA (%), and water holding capacity as compared to other nutrient management. GreenSeeker-based nutrition ensured the best N-management in soil and there is a strong relationship between soil N and SOC [28]. Thus, this nutrient management ensured better soil SOC resulting in better soil physical properties than other management.

Under long-term CA (conservation agriculture)-based practices viz. PB and ZT/direct seeding system, the SOC build-up was found to be ~30% more than the SOC under CT. In PB and ZT, the soil was disturbed less, and crop residues were maintained on topsoil resulting in less oxidation of soil organic matter and more organic matter build-up after residue decomposition [3,15]. On the other hand, in CT, continuous soil disturbance facilitated the oxidation of the soil organic matter resulting in less SOC [34,35]. Precision N-management using GreenSeeker or Nutrient Expert Software resulted in ~10% more SOC comparing the SOC under FFP. Application of N-fertilizers was higher under FFP in maize crops comparing SSNM using Nutrient Expert Software, and GreenSeeker-based N-management, while the latter two nutrient management practices focused on balanced nutrition. Jin [36] and Laik et al. [28] stated that the soil organic-C distribution can be rearranged through mineral N-fertilizer application, and the use of optimal N-fertilization can enhance the SOC, while indiscriminate N-application reduces the SOC.

Conservation tillage also improved other soil chemical properties as compared to CT. Continuous tillage operation without maintaining crop residues might result in the shifting of the fertile topsoil with less fertile subsoil during soil disturbance. Such inversion might cause possible leaching losses of soil nutrients [37]. Many previous research findings also showed that in conservation tillage practices, soil chemical properties were improved due to higher SOC build-up in soil [38–41]. A higher correlation of SOC with other soil chemical properties in this study as depicted in Table 4 also confirmed this finding.

**Table 4.** Correlation between SOC and other soil chemical and biological properties.

|  | SOC | N | P | K | Zn | Fe | AC | MBC | DHA |
|---|---|---|---|---|---|---|---|---|---|
| SOC | 1 | | | | | | | | |
| N | 0.995 ** | 1 | | | | | | | |
| P | 0.998 * | 0.997 * | 1 | | | | | | |
| K | 0.986 | 0.998 * | 0.999 * | 1 | | | | | |
| Zn | 0.999 * | 0.999 * | 0.999 * | 0895 | 1 | | | | |
| Fe | 0.889 | 0.991 | 0.894 | 0.998 * | 0.886 | 1 | | | |
| AC | 0.996 * | 0.998 * | 0.991 * | 0.919 ** | 0.996 | 0.997 * | 1 | | |
| MBC | 0.997 * | 0.999 ** | 0.995 ** | 0.999 * | 0.999 * | 0.893 | 0.999 * | 1 | |
| DHA | 0.995 | 0.907 | 0.998 * | 0.998 ** | 0.994 | 0.999 * | 0.998 * | 0.998 * | 1 |

* Correlation is significant at the 0.05 level (2-tailed); ** Correlation is significant at the 0.01 level (2-tailed) [SOC: soil organic carbon; N: soil available nitrogen; P: soil available phosphorus; K: soil available potassium; Zn: zinc; Fe: iron; AC: active carbon; MBC: microbial biomass carbon; DHA: dehydrogenase activity].

Soil available N and K were significantly influenced by different nutrient management practices. FFP showing a slightly higher available N compared to the other nutrient management might be attributed to the application of the higher amount of external chemical N-fertilizer. Moreover, the available N in soil not being so high in FFP as compared to the other two precision nutrient management practices might be due to the loss of excess N through leaching. In SSNM and GreenSeeker-based treatment, N-fertilizer was applied as per the crop demand and target yield which ensured optimal N-management resulting in less N-loss [13,42]. Soil available K was found to be the maximum under SSNM being at par with GreenSeeker-based treatment. A higher amount of available K in SSNM was again attributed to the application of the higher number of external K-fertilizers compared to the other two treatments.

Soil biological properties were also found to be significantly varied with different tillage practices. PB and ZT/direct seeding resulted in better soil AC, MBC, and DHA, respectively, as compared to those under CT. Higher soil AC, MBC, and DHA might be attributed to the higher amount of soil SOC under PB and ZT. Higher SOC in soil confirmed the higher microbial activity which ultimately resulted in higher soil AC and MBC [3,28,43,44]. A strong relationship between AC with SOC; MBC with SOC; and DHA with MBC depicted in Equations (4)–(6), respectively also confirmed the above statement.

$$AC = 0.136SOC + 0.186; R^2 = 0.983 \tag{4}$$

$$MBC = 0.019SOC - 2.25; R^2 = 0.993 \tag{5}$$

$$DHA = 8.144MBC + 188.3; R^2 = 0.988 \tag{6}$$

Precision nutrient management using Nutrient Expert Software or GreenSeeker-based reading showed better soil biological properties over farmers' practice. Such better soil biological properties under precision nutrient management over FFP were again attributed to the better SOC and soil aggregation under these treatments comparing FFP. Both precision nutrient management practices focused on balanced nutrition for the crops resulting in better soil health. Thus, the soil's biological properties were improved under these treatments. This result also confirms the previous results recorded by Parihar et al. [13,15].

Both conservation agricultural practices resulted in higher system production in terms of MEY as compared to that under CT. Such higher system production might be attributed to the better soil physical, chemical, and biological properties under conservation tillage practices in the maize–wheat system. Improved soil bio-physico-chemical properties under PB and ZT/direct seeding ensured better nutrient availability to the plant at the critical crop–nutrient demand, thereby, overall growth and development of the plants were conducted, and this ultimately resulted in higher yield [3,45–47]. Similarly, the precision nutrition to the system resulted in higher system production compared to that under CT. Both the precision nutrient management practices ensured optimal nutrient supply to the crop as per their demand resulting in better biomass production. Under FFP, N-fertilizer was applied in more amounts without considering the critical demand of the crop and soil response. As a result, additional N was not utilized by the crop resulting in comparatively poor yield than less but balanced and well-managed N along with other nutrients viz. P and K applied SSNM and GreenSeeker plots [48].

Conservation agriculture practice viz. PB and ZT/direct seeding showed an additional net return of about US$330–US$400 over the net return under CT. Such additional net return as well as the benefit–cost ratio under conservation tillage practices might be attributed to higher system production than those under CT, and less operative cost under conservation tillage due to saving in the land preparation because of minimum soil disturbance [49]. Smart nutrient management using Nutrient Expert Software or GreenSeeker exhibited a higher net return and B–C ratio as compared to those under farmers' practice. Such improvement in profits under SSNM and GreenSeeker plots might be attributed to higher yield with a balanced nutrient supply as per the demand of the crops. Kapri and Kesar-

wani [50]; Kumar et al. [51] also demonstrated that precision nutrient management can improve the economics of crop cultivation.

## 5. Conclusions

From this long-term experiment, it was observed that the use of conservation tillage viz. permanent bed system (PB) or zero tillage (ZT)/direct seeding, plus the retention of previous-crop residues, can improve soil organic carbon (SOC) by 31–33% over conventional tillage (CT) with no residue retention. Such an increase in SOC helped in a decrease in BD, an increase in WSA (%), and the water holding capacity of the soil to a considerable amount over CT. These tillage systems also improved the soil chemical as well as biological status as compared to those under CT. Thereby, both these conservation agriculture practices augmented the maize–wheat system yield by about 13–18% over CT, and showed an additional net return of US$330–US$400 over CT. Comparing PB and ZT, it was observed that PB was a little bit better concerning the improvement of soil properties, yield, and economics under the maize–wheat cropping system. SSNM using Nutrient Expert® software or GreenSeeker-based nutrient management resulted in better soil biological and chemical properties over the farmers' practice. Both these precision nutrient management practices resulted in significantly higher system yield and economics as compared to farmers' practice. Comparing SSNM and GreenSeeker, it was observed that the SSNM was a bit better concerning the improvement of system yield and economics. Thus, from this study, it can be recommended that permanent bed planting with residue retention was the best conservation tillage practice concerning the improvement of soil properties, system yield, and profits under the maize–wheat system. Concerning nutrient management, site-specific nutrient management using Nutrient Expert® software was found to be the best nutrition option for improving the soil chemical and biological properties, thereby, improving the system yield and economics of the maize–wheat cropping system.

**Supplementary Materials:** The following supporting information can be downloaded at: https://www.mdpi.com/article/10.3390/agronomy12112766/s1, Supplementary Table S1: Cost of cultivation for the maize–wheat cropping system in US$.

**Author Contributions:** Conceptualization, M.K. (Mritunjay Kumar) and B.P.; methodology, M.K. (Mukesh Kumar) and B.P.; software, B.P., B.S.S.S.N. and V.D.R.; validation, M.K. (Mritunjay Kumar), B.M.N., M.K. (Mukesh Kumar) and S.M.; formal analysis, B.M.N., S.K.S. and B.P.; investigation, B.M.N., B.P., S.K.S. and M.K. (Mritunjay Kumar); resources, M.K. (Mritunjay Kumar); data curation, B.P., V.D.R. and S.M.; writing—original draft preparation, B.P.; writing—review and editing, S.M., V.D.R., T.M. and B.S.S.S.N.; visualization, M.K. (Mritunjay Kumar); supervision, M.K. (Mukesh Kumar) and B.P.; project administration, M.K. (Mritunjay Kumar) and B.P.; funding acquisition, M.K. (Mritunjay Kumar), B.P., V.D.R. and T.M. All authors have read and agreed to the published version of the manuscript.

**Funding:** The study was funded by the ICAR-IIMR, Ludhiana, Director Research, RPCAU, and Pusa for providing essential financial and other assistance to conduct this long-term study.

**Data Availability Statement:** All the data are incorporated in the tables and figures.

**Acknowledgments:** All the authors duly acknowledge the research team of maize agronomy of RPCAU, Pusa for the successful conduction of this long-term study.

**Conflicts of Interest:** All authors declared no conflict of interest.

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
