# Peer review of "Long-Term Conservation Tillage and Precision Nutrient Management in Maize–Wheat Cropping System: Effect on Soil Properties, Crop Production, and Economics"

_agronomy, doi:10.3390/agronomy12112766_

Round 1
Reviewer 1 Report
This study investigated on impact of conservation tillage and precision nutrient management options on soil health, system yield, and economic benefits of maize–wheat system. It's a seven-years field study and the results are valuable. But, in spite of having huge data, the paper is faced with lack of good discussion. Anyway I think It's potential to be publish after passing a major revision. It's mainly about discussion. Other sections are acceptable. please see comments below.
Line 38: Please make your keywords different from title words. Most of them are repeated from title!
Line 113: Please attention to superscript and subscript numbers in whole text. for ex: CaCO3.
Table 2: Tillage practice in both plants are same. So, remove last column and change table's title to: Description of tillage practices in cultivation.
There is an important point about results and discussion. Based on journal's instructure, results and discussion should be separate into two sections, but here you combined them in one section. Besides, even the presented discussion is NOT enough and need to be more extent. This is a seven years study, so It's worth to spent more time on interpretation of findings and compare your findings with other studies. Currently, in most of parts related to discussion, you just mentioned:''Similar findings were recorded by [00]''. It's NOT discussion!
Lines 270-274: Remove this sentences from results and place it in discussion section! Because here you just need to report your findings. Interpretation, why, how, ... should be present in discussion.
Figures are NOT clear! They SHOULD be replaced with high resolution version.
Author Response
Response to Reviewer 1 Comments
Point 1: Line 38: Please make your keywords different from title words. Most of them are repeated from title!
Response 1: As per the suggestion of the respected reviewer, we have changed the keywords avoiding the repetition of the same words mentioned in the title.
Point 2: Line 113: Please attention to superscript and subscript numbers in whole text. for ex: CaCO3.
Response 2: We have changed it as per the suggestion of the respected reviewer.
Point 3: Table 2: Tillage practice in both plants are same. So, remove last column and change table's title to: Description of tillage practices in cultivation.
Response 3: We agree with the reviewer’s comment, however, Table 2 contains both tillage and nutrient management details. Tillage practices for maize and wheat were the same, however, the nutrient levels were not the same. So, we put both maize and wheat in the column.
Point 4: L There is an important point about results and discussion. Based on journal's instructure, results and discussion should be separate into two sections, but here you combined them in one section. Besides, even the presented discussion is NOT enough and need to be more extent. This is a seven years study, so It's worth to spent more time on interpretation of findings and compare your findings with other studies. Currently, in most of parts related to discussion, you just mentioned:''Similar findings were recorded by [00]''. It's NOT discussion!
Response 4: We have separated the results and discussion parts and improved the discussion part as mentioned by the respected reviewer.
Point 5: Lines 270-274: Remove this sentences from results and place it in discussion section! Because here you just need to report your findings. Interpretation, why, how, ... should be present in discussion.
Response 5: We have moved all these discussion parts from the results section to the discussion section.
Point 6: Figures are NOT clear! They SHOULD be replaced with high resolution version.
Response 6: We have incorporated the higher-resolution images. All the images are 600 dpi resolution now.

Reviewer 2 Report
Respected Sir,
Thanks for considering me to review the manuscript titled “Low-carbon calcareous soils can be productive and economic for maize–wheat cropping systems with long-term conservation tillage and precision nutrient management”
As the study has some novelty, so it should be improved and published. The manuscript needs some minor revisions addressed. The MS must be revised to avoid the typos mistakes.
Abstract: Well written
Introduction covered the research point from all sides. Please improve the aim of the study and raise the novelty. Improve the quality of the figures.
Methods: The description of analysis methods are written in a short way and they need to more descriptive to enable other to repeat the experiment.
Results: The description of results is satisfied.
Discussion: Results should be justified accordingly with some latest references and modern style.
Overall, the manuscript should be improved with some latest references and needs some formatting as the manuscript has been opened in different MS office.
Conclusion: Is well written but it needs to be more focus on the practical results of the study.
Author Response
Response to Reviewer 2 Comments
Point 1: Abstract: Well written
Response 1: Thank you very much for your comment.
Point 2: Introduction covered the research point from all sides. Please improve the aim of the study and raise the novelty. Improve the quality of the figures.
Response 2: We have edited the aim of the study and included the novelty statement as per the comment of the respected reviewer. We also improve the quality of the figures. All the figures are of 600 dpi resolution now.
Point 3: Methods: The description of analysis methods are written in a short way and they need to more descriptive to enable other to repeat the experiment.
Response 3: : As per the comment of the respected reviewer, we have included a bit of detail about the analytical procedure. We already described details of the soil chemical procedure and microbial biomass carbon in our previously published papers which are mentioned by citing the reference in this article. We avoid repeating that to reduce the plagiarism/ similarity index of this paper. We have included the details of other analytical procedures as suggested by the reviewer.
Point 4: Results: The description of results is satisfied
Response 4: Thank you for your comment.
Point 5: Discussion: Results should be justified accordingly with some latest references and modern style.
Response 5: We have separated the discussion part from the results section and improved it as per the suggestions of the respected reviewer.
Point 6: Overall, the manuscript should be improved with some latest references and needs some formatting as the manuscript has been opened in different MS office.
Response 6: We have followed this instruction of the respected reviewer.
Point 7: Conclusion: Is well written but it needs to be more focus on the practical results of the study.
Response 7: We have changed the conclusion part as per the suggestion of the respected reviewer.

Reviewer 3 Report
Title.
In my opinion, the title should be changed. The title cannot be a conclusion or statement before reading the manuscript.
It is not necessary to enter "calcareous soils" in the title. The soil type may be entered in keywords and in the test method. The more that it is not a factor of experience.
Abstract.
It is corectly described
Key word:
Corectly choosen
Introduction.
It is corectly described
Materials and methods.
The chapter is generally well described. Such changes should be taken into account:
- instead of "Zero tillage", authors should use the term: "Direct seeding". This term has not been used for a long time because it is not precise. Merely performing sowing and introducing seeds is a kind of tillage.
- Organic carbon is a chemical property. It should be described (also in the Results chapter) together with the chemical properties.
Line 193. - The statement "Afterward, samples were air-dried and processed to analyze the following parameters:" and the continuation of the subchapters is quite unfortunate here.
Statistical analysis.
Basically correctly done. However, my doubts are why the tillage * and precision nutrient management cooperation was not done. Please explain because these could be interesting results.
Results.
Line 264 - 3. Results. It should read "Results and Discussion
Table / Figure. The expansion of experience factors abbreviations should be inserted under the tables / figures.
Line 328. - 3.3 Soil chemical properties
Line 328. - 3.3 Soil chemical properties
1) This chapter includes a table with correlations data on biological properties, not just chemical properties,
3) Explanation of abbreviations should be inserted
4) Performing correlations without indicating what these were the correlations under the influence of the examined factors seems to me unnecessary. Currently, these data are quite obvious.
Conclusion
Line 460. Due to the fact that the publication does not present the interaction of experimental factors, and only the main effects, I believe that the statement: "Thus, from this study, it can be recommended that permanent bed planting with residue retention coupled with site-specific nutrient management using Nutrient Expert® software can improve the soil-physical, chemical, and biological properties, thereby, improving the system yield and economics of maize – wheat cropping system "is unauthorized
Author Response
Response to Reviewer 3 Comments
Point 1: Title: In my opinion, the title should be changed. The title cannot be a conclusion or statement before reading the manuscript.
It is not necessary to enter "calcareous soils" in the title. The soil type may be entered in keywords and in the test method. The more that it is not a factor of experience.
Response 1: We have changed the title as suggested by the respected reviewer. We also deleted the “calcareous soils” from the title and put that into the keywords.
Point 2: Abstract: It is correctly described
Response 2: Thanks for your comment.
Point 3: Key word: Corectly choosen
Response 3: Thanks for your comment.
Point 4: Introduction: It is corectly described
Response 4: Thanks for your comment.
Point 5: Materials and methods. The chapter is generally well described. Such changes should be taken into account: - instead of "Zero tillage", authors should use the term: "Direct seeding". This term has not been used for a long time because it is not precise. Merely performing sowing and introducing seeds is a kind of tillage.
Response 5: We use the term “Zero tillage” being an important part of conservation agriculture. We appreciate the comment of the respected reviewer to use the term “Direct seeding.” So, we use both terms whenever it was applicable.
Point 6: Organic carbon is a chemical property. It should be described (also in the Results chapter) together with the chemical properties.
Response 6: We agree with the respected reviewer. We include the organic carbon section in the soil chemical analysis part.
Point 7: Line 193. - The statement "Afterward, samples were air-dried and processed to analyze the following parameters:" and the continuation of the subchapters is quite unfortunate here.
Response 7: We completed the sentence and then we started the sub-chapters as suggested.
Point 8: Statistical analysis: Basically correctly done. However, my doubts are why the tillage * and precision nutrient management cooperation was not done. Please explain because these could be interesting results.
Response 8: We could not include the tillage * nutrient management data as we observed that the analysed data were statistically non-significant. So, we avoid including those data in this article.
Point 9: Results: Line 264 - 3. Results. It should read "Results and Discussion
Response 9: We have separated the results and discussion part as per the other reviewer’s comments and the journal’s requirements.
Point 10: Table / Figure. The expansion of experience factors abbreviations should be inserted under the tables / figures.
Response 10: We have added all the abbreviations in the footnote of the tables and figures.
Point 11: Line 328. - 3.3 Soil chemical properties
1) This chapter includes a table with correlations data on biological properties, not just chemical properties,
Response 11: We include the correlation matrix for better discussion to demonstrate the relationship between SOC with soil chemical properties. So that we can show improvement in the SOC under conservation tillage or precision nutrient management can improve the soil chemical properties also.
Point 12: 3) Explanation of abbreviations should be inserted
Response 12: We included the full form of the abbreviation in the tables and figures as well as in the text whenever those appear first.
Point 13: 4) Performing correlations without indicating what these were the correlations under the influence of the examined factors seems to me unnecessary. Currently, these data are quite obvious.
Response 13: We appreciate the comments of the respected reviewer. We included those correlations for a better discussion of the results obtained in the study.
Point 14: Conclusion: Line 460. Due to the fact that the publication does not present the interaction of experimental factors, and only the main effects, I believe that the statement: "Thus, from this study, it can be recommended that permanent bed planting with residue retention coupled with site-specific nutrient management using Nutrient Expert® software can improve the soil-physical, chemical, and biological properties, thereby, improving the system yield and economics of maize – wheat cropping system "is unauthorized
Response 14: We agree with the comment of the respected reviewer. We have rephrased the conclusion part accordingly.

Round 2
Reviewer 1 Report
The Authors improved the manuscript and it potentially can be publish.